# Tyrosine Is a Booster of Leucine-Induced Muscle Anabolic Response

**DOI:** 10.3390/nu16010084

**Published:** 2023-12-26

**Authors:** Kotaro Tamura, Hidefumi Kitazawa, Satoshi Sugita, Kohjiro Hashizume, Masazumi Iwashita, Takaaki Ishigami, Yoshihiko Minegishi, Akira Shimotoyodome, Noriyasu Ota

**Affiliations:** 1Biological Science Research, Kao Corporation, 2606 Akabane, Ichikai-machi, Haga-gun, Tochigi 321-3497, Japan; tamura.kotaro@kao.com (K.T.); sugita.satoshi@kao.com (S.S.);; 2Processing Development Research, Kao Corporation, Wakayama-shi, Wakayama 640-8580, Japan

**Keywords:** amino acids, mTOR complex 1, muscle cells, Sestrin1/2

## Abstract

Leucine (Leu), an essential amino acid, is known to stimulate protein synthesis in the skeletal muscle via mTOR complex 1 (mTORC1) activation. However, the intrinsic contribution of other amino acids to Leu-mediated activation of mTORC1 signaling remains unexplored. This study aimed to identify amino acids that can promote mTORC1 activity in combination with Leu and to assess the effectiveness of these combinations in vitro and in vivo. We found that tyrosine (Tyr) enhanced Leu-induced phosphorylation of S6 kinase (S6K), an indicator of mTORC1 activity, although it exerted no such effect individually. This booster effect was observed in C2C12 cells, isolated murine muscle, and the skeletal muscles of mice orally administered the amino acids. To explore the molecular mechanisms underlying this Tyr-mediated booster effect, the expression of the intracellular Leu sensors, Sestrin1 and 2, was suppressed, and the cells were treated with Leu and Tyr. This suppression enabled Tyr alone to induce S6K phosphorylation and enhanced the booster effect, suggesting that Tyr possibly contributes to mTORC1 activation when Sestrin-GAP activity toward Rags 2 (GATOR2) is dissociated through Sestrin knockdown or the binding of Sestrins to Leu. Collectively, these results indicate that Tyr is a key regulator of Leu-mediated protein synthesis.

## 1. Introduction

Skeletal muscle mass is maintained through a balance between protein synthesis and degradation. Nutritional supplementation with proteins or amino acids activates anabolic responses in the skeletal muscle and may be important for counteracting muscle loss due to aging, sarcopenia, or frailty [1,2]. The intracellular signaling mechanism regulating muscle protein synthesis (MPS) is controlled by the activation of the mammalian target of rapamycin complex 1 (mTORC1), which directly stimulates the phosphorylation of S6 kinase (S6K) and eukaryotic translation initiation factor 4E-binding protein (4E-BP). Changes in the phosphorylation state of these key proteins affect mRNA translation initiation and elongation, thereby regulating MPS [3]. Essential amino acid (EAA) supplementation effectively stimulates MPS; however, non-essential amino acids (NEAAs) are ineffective even at significantly high doses [4,5]. Among EAAs, leucine (Leu) has been shown to be particularly important for MPS, as it is the only stimulator of mTORC1 signaling identified in muscle cells over the physiological range of amino acid levels in blood [6]. The amount of Leu in ingested proteins or EAA mixtures determines the extent of the MPS response at rest and after exercise [7,8,9,10,11]. The amino acid composition of whey protein is considered to be suitable for stimulating MPS, owing to its high Leu content and absorbability [12,13]. Thus, Leu is widely accepted as being indispensable for stimulating mTORC1 signaling. Recently, the mechanism underlying Leu-mediated mTORC1 activation was elucidated using HEK293T cells [14,15]. In these studies, mTORC1 regulation by amino acids was found to be mediated by Rag guanosine triphosphatase (GTPase) via several factors, including GATOR1, a GTPase-activating protein that inhibits mTORC1 in response to amino acid starvation, GATOR2, which inhibits GATOR1 activity, and Sestrin2, a GATOR2-interacting protein [16,17]. Leu disrupts the Sestrin2–GATOR2 interaction by binding to Sestrin2, and GATOR2 in turn activates mTORC1 by binding to GATOR1 [14,15]. Although three Sestrin isoforms, 1–3, are expressed in mammalian cells, Leu promotes only the dissociation of Sestrin1 and Sestrin2 from GATOR2 [14]. In skeletal muscle, Leu-mediated mTORC1 activation occurs primarily through Sestrin1 rather than Sestrin2, as Sestrin1 is more highly expressed than Sestrin2 [18].

Though Leu is undoubtedly important for MPS, recent studies have revealed several mechanisms related to other EAAs, including glutamine [19], arginine [20], and methionine [21], underlying mTORC1 activation. However, to the best of our knowledge, the intrinsic contribution of other amino acids, including NEAAs, to Leu-mediated activation of mTORC1 signaling has not yet been assessed. This study aimed to identify amino acids that play a role in enhancing Leu-mediated activation of mTORC1 signaling in skeletal muscle. Collectively, our data provide novel insights into the most appropriate composition of daily protein/amino acid macronutrients or supplements.

## 2. Materials and Methods

### 2.1. Cell Culture and Treatments

C2C12 myoblasts (European Collection of Authenticated Cell Cultures, Salisbury, UK) were seeded at a density of 1.25 × 10^5^ cells/well in 8-well rectangular plates and maintained in Dulbecco’s Modified Eagle’s medium (DMEM; 25 mM glucose; Sigma, St. Louis, MO, USA) supplemented with 10% fetal bovine serum (FBS; Biosera, Kansas City, MO, USA) and 1% penicillin–streptomycin (PS; Life Technologies, Carlsbad, CA, USA) at 37 °C in a 5% CO_2_-containing atmosphere. At 80% confluence, the culture medium was changed to a differentiation medium consisting of DMEM supplemented with 2% horse serum (HS; Life Technologies) and 1% PS (day 0). The cells were collected and used for western blotting and the evaluation of myotube hypertrophy on days 1 and 5 post-differentiation, respectively. The cells were deprived of serum and amino acids through incubation in Hanks’ Balanced Salt Solution (HBSS with Ca and Mg, without Phenol Red; Life Technologies, Carlsbad, CA, USA) for 3–4 h; this was followed by inoculation with various amino acids for 10–30 min. Amino acids used for the cell experiments were purchased from Fujifilm Wako (Osaka, Japan).

### 2.2. siRNA Transfection

To deliver oligonucleotide-based siRNAs (Silencer Select pre-designed siRNA; Thermo Fisher Scientific, Waltham, MA, USA) into cells, 50–60% confluent C2C12 cells were transfected with 10 nM siRNA using lipofectamine RNAiMAX (Invitrogen, Carlsbad, CA, USA) and Opti-MEM I Reduced-Serum Medium (Invitrogen, Carlsbad, CA, USA), according to the instructions of the manufacturer. The culture medium was changed to a differentiation medium 24 h after transfection, and the cells were collected and used for experiments on day 1 post-differentiation. The siRNAs used in this study were the Silencer Negative Control #2, Sestrin1 (s100521), Sestrin2 (s100521), and Leucyl-tRNA Synthetase (LRS, s98763) siRNAs.

### 2.3. Evaluation of Amino Acid-Induced S6K Phoshorylation

C2C12 myoblasts were seeded at a density of 6 × 10^4^ cells/well in 96-well plates and cultured at 37 °C and in a 5% CO_2_-containing environment overnight. The following day, the medium was replaced with HBSS, and the cells were cultured for 3–4 h. Subsequently, the cells were incubated with various concentrations of amino acids for 15 min. After removal of HBSS, the cells were lysed using CelLytic MT Cell Lysis Reagent (C3228; Sigma, St. Louis, MO, USA) for 10 min, and the extract was subjected to an AlphaScreen SureFire assay (TGR70S500; PerkinElmer, Waltham, MA, USA) to detect phospho-p70 S6K (Thr 389) following the instructions of the manufacturer. The values were corrected for protein concentrations determined separately using a Pierce BCA Protein Assay Kit (Thermo Fisher Scientific, Waltham, MA, USA). All data are presented relative to those of control-deprived amino acids.

### 2.4. Immunohistochemistry and Evaluation of Myotube Hypertrophy

C2C12 myoblasts were cultured for five days in a differentiation medium supplemented with Leu and/or tyrosine (Tyr). During this period, the medium was changed daily. On day 5 post-differentiation, C2C12 myotubes were washed with PBS and fixed with 4% paraformaldehyde PBS (Fujifilm-Wako, Osaka, Japan) for 20 min at 4 °C. The fixed cells were washed thrice with PBS and incubated with 0.2% TritonX-100 (Sigma, St. Louis, MO, USA) in PBS for 10 min at room temperature (RT). After incubation in a blocking solution (3% BSA, Sigma, St. Louis, MO, USA; in PBS) for 30 min at RT, the cells were incubated with primary antibodies against the Myosin heavy chain (MyHC, MF 20sp; DSHB, Iowa City, IA, USA), diluted in the blocking solution to 5 µg/mL, at 4 °C overnight. After the cells were washed in PBS, they were incubated with secondary antibodies (Alexa 488-conjugated donkey anti-mouse antibodies (1:500, Molecular Probes, Thermo Fisher Scientific, Waltham, MA, USA)) diluted in the blocking solution for 1 h at RT. The cells were subsequently washed with PBS and stained with DAPI (Dojindo, Kumamoto, Japan). Images of the stained cells were captured using an all-in-one fluorescence microscope (BZ-X710; Keyence, Osaka, Japan) at a magnification of 10×. The average myotube diameter was calculated as the mean of three measurements taken along the long axis of the myotubes using a BZ Analyzer (Keyence); in total, 133–142 myotubes were evaluated from four random fields.

### 2.5. Animals and Diets

Male C57BL/6J mice were purchased from CLEA Japan (Tokyo, Japan) and maintained under controlled conditions (temperature: 23 ± 2 °C; humidity: 55 ± 10%; lighting: 07:00 to 19:00 h). The mice were provided standard chow (CE-2; CLEA Japan, Tokyo, Japan) and ad libitum access to water. All animal experiments were conducted at the Experimental Animal Facility of the Kao Tochigi Institute (Tochigi, Japan) and were approved by the Animal Care Committee of the Kao Corporation (Tokyo, Japan). Food additive-grade L-Leu and L-Tyr used in the animal experiments were purchased from Ajinomoto Healthy Supply (Tokyo, Japan).

#### 2.5.1. Animal Experiment 1: Incubation of Isolated Muscles

Overnight-fasted mice (14 weeks old, *n* = 20) were sacrificed, and their soleus and the extensor digitorum longus (EDL) muscles were removed from both hindlimbs. Two muscles isolated from both legs of one mouse were independently used in the experiment. The isolated soleus and EDL muscles (*n* = 40 each) were randomly assigned to one of six treatment groups. The incubation of the isolated muscles was performed as previously described [22], with slight modifications. Briefly, isolated muscles tied with silk thread at both tendon ends were mounted on an incubation apparatus and pre-incubated in Krebs Ringer Bicarbonate (KRB) buffer (K4002, Sigma, St. Louis, MO, USA) for 10 min. Thereafter, the buffer was replaced with a fresh KRB buffer supplemented with Leu and/or Tyr. In addition, 95% O_2_-5% CO_2_ was continuously bubbled through the buffer at 37 °C. After 20 min of incubation, the isolated muscles were washed with ice-cold KRB buffer and stored at −80 °C until further analyses.

#### 2.5.2. Animal Experiment 2: Oral Administration of Amino Acids

Overnight-fasted mice (8 weeks old, *n* = 51) with similar average body weights were randomly assigned to one of the seven dietary treatments. The control group received an emulsion containing 2 g of fat per kg of body weight (BW), and the treatment groups received the same amount of emulsion with either Leu at 5 mmol/kg BW, Leu at 10 mmol/kg BW, Tyr at 1 mmol/kg BW, Tyr at 5 mmol/kg BW, Leu at 5 mmol/kg BW + Tyr at 1 mmol/kg BW, or Leu at 5 mmol/kg BW + Tyr at 5 mmol/kg BW. The maximum amount of Leu (10 mmol/kg BW), which was the equivalent of the amount of Leu consumed by rats per day, was determined based on data from previous studies [23,24]. Glyceryl trioleate (Sigma, St. Louis, MO, USA) was used as a fat source. Lecithin from egg yolk (Kanto Chemical, Tokyo, Japan) was added to all test emulsions at 0.08 g/kg BW (0.4% (*w*/*w*)) in the administered samples. Premixed solutions were subsequently sonicated thrice for 60 s, with a 1 min interval of cooling on ice, to obtain stable emulsions (Sonifier 450; Branson Ultrasonics, Danbury, CT, USA), as previously described [25]. The lipid emulsion was intragastically administered to mice under isoflurane anesthesia at a dose of 20 mL/kg BW (Abbott Laboratories, Chicago, IL, USA), and then the mice were returned to their cages. Thirty minutes after amino acid administration, blood samples were collected from the abdominal venae cavae of the mice under isoflurane anesthesia, and the mice were sacrificed. The muscles were collected from both mouse hindlimbs and stored at −80 °C until analysis. The collected blood samples were preserved on ice and centrifuged at 10,000× *g* for 6 min at 4 °C, and mouse sera were stored at −80 °C until analysis.

### 2.6. Western Blotting

Cells or frozen muscle samples were homogenized in the CelLytic MT Cell Lysis Reagent (Sigma, St. Louis, MO, USA) supplemented with a complete protease inhibitor cocktail (Roche, Basel, Switzerland) and a phosphatase inhibitor cocktail solution (Fujifilm-Wako) using an ultrasonic homogenizer (Sonifier 150; Branson Ultrasonics, Danbury, CT, USA) or a handy micro homogenizer (PHYSCOTRON, NS-310E3; MICROTEC, Chiba, Japan), respectively. After centrifugation at 13,500× *g* for 15 min at 4 °C, the protein concentration of the supernatant was determined using the Pierce BCA Protein Assay Kit. Samples were separated on an SDS-polyacrylamide gel (10% or 4–15% Mini-PROTEAN TGX Gels; Bio-Rad, Hercules, CA, USA) and transferred onto polyvinylidene fluoride (PVDF) membranes (ClearTrans SP; Fujifilm-Wako). The membranes were then blocked with a PVDF blocking reagent (TOYOBO, Osaka, Japan). Subsequently, the membranes were incubated overnight with anti-S6K (9202; Cell Signaling Technology (CST), Danvers, MA, USA), anti-P-S6K (Thr389, 9205; CST), anti-4E-BP (9452; CST), anti-P-4E-BP (Thr37/46, 2855; CST), anti-mTOR (2972; CST), anti-P-mTOR (Ser2448, 2971; CST), anti-P-S6 (Ser235/236, 2211; Ser240/244, 2215; CST), anti-S6 (2217; CST), anti-GAPDH (2118, CST), anti-LRS (13868, CST), anti-Sestrin1 (21668-1-AP, Proteintech, Rosemont, IL, USA), and anti-Sestrin2 (10795-1-AP, Proteintech) antibodies, diluted at a 1:2000 ratio using an immunoreaction-enhancing solution (Can Get Signal, TOYOBO). The appropriate secondary antibodies conjugated with horseradish peroxidase (7074; CST, dilution 1:2000) were used to detect blots via enhanced chemiluminescence on the ECL Prime (GE Healthcare, Chicago, IL, USA). Images of the blots were captured using a ChemiDoc MP Imaging System (Bio-Rad, Hercules, CA, USA), and band volumes were adjusted with local background subtraction using the Image Lab software (ver. 6.0.1, Bio-Rad, Hercules, CA, USA).

### 2.7. Muscle Protein Synthesis

Muscle protein synthesis was evaluated using the surface sensing of translation (SUnSET) method [26,27]. C2C12 myoblasts (day 1) were incubated with 1 μM puromycin (Abcam, Cambridge, UK) and amino acids in HBSS buffer at 37 °C for 30 min. Then, the cells were collected and subjected to western blotting following the method described above. Anti-puromycin antibodies (MABE343; Millipore, Burlington, MA, USA) were used to detect puromycin incorporation in de novo proteins. The intensities of all puromycin-labeled protein bands were normalized to that of Coomassie blue staining (Bio-Safe G-250; Bio-Rad) in each lane.

### 2.8. Measurement of Amino Acid Levels

The serum or muscle samples were homogenized using three times the volume of 15% sulfosalicylic acid (Fujifilm-Wako), and the homogenate was centrifuged at 10,000× *g* for 10 min at 4 °C to remove proteins. Then, the supernatant was subjected to subsequent analyses. Free Leu and Tyr levels in the samples were determined using an incorporated LC-MS/MS system (Infinity 1290; Agilent, Santa Clara, CA, USA, and the QTRAP system; AB Sciex, Tokyo, Japan) with a Scherzo SS-C18 column (Imtakt, Kyoto, Japan). Mobile phase A consisted of 0.1 M ammonium formate in Milli-Q water, and mobile phase B consisted of 0.3% formic acid in MeOH. The initial eluent was composed of 1% B, followed by a linear increase to 90% B within 3 min. This proportion was maintained for 1 min; then, the mobile phase was returned to the initial condition and maintained for 1 min until the end of the run. The total running time was 5 min, eluent flow was 1 mL/min, and column temperature was set at 40 °C. Mass spectroscopic detection and quantification of the analytes were performed using a multiple-reaction-monitoring (MRM) scan device in positive ion mode. Q1 and product ion scans were acquired by infusing the sample solutions of each analyte with a mixed mobile phase solution using an infusion pump. The source temperature and gas parameters were optimized after the chromatographic conditions were fixed as follows: curtain gas: 20 psi; collision gas: 8 psi; ion spray voltage: 5000 V; source temperature: 600 °C; ion source gas 1: 65 psi; ion source gas 2: 30 psi. The ionization parameters of the analytes were as follows: for Leu: declustering potential (DP): 51 V; entrance potential (EP): 10 V; collision energy (CE): 19 V; and collision cell exit energy (CXE): 36 V; for Tyr: DP: 56 V; EP: 10 V; CE: 15 V; and CXE: 22 V. The analytes were detected in a multiple-reaction-monitoring mode by monitoring the characteristic fragmentation ions (*m*/*z* 132.14 > 86.3) for Leu and (*m*/*z* 182.11 > 91.1) for Tyr.

### 2.9. RNA Extraction and RT-PCR

Total RNA was extracted from C2C12 cells using an RNeasy Mini Kit (Qiagen, Hilden, Germany), following the protocol of the manufacturer. RNA was transcribed into cDNA using a high-capacity RNA-to-cDNA Kit (Applied Biosystems, Foster City, CA, USA). Quantitative RT-PCR was performed using the ABI Prism 7500 device with TaqMan gene expression assays (Applied Biosystems, Foster City, CA, USA). The mRNA levels of each gene were normalized to the average of those of two housekeeping genes, 18S ribosomal RNA (18S rRNA) and glyceraldehyde 3-phosphate dehydrogenase (GAPDH). The probes used in this study included Mm01185732_m1 for Sestrin1, Mm00460679_m1 for Sestrin2, Mm03928990 for 18S rRNA, and Mm99999915_g1 for GAPDH.

### 2.10. Statistical Analysis

All data are presented as mean ± standard error (SE). Differences between multiple groups were tested using one-way ANOVA followed by Tukey’s post-hoc test. Comparison of two factors was performed using two-way ANOVA followed by Tukey’s post hoc test when a significant main effect or interaction was observed. Correlation analysis was performed using the Pearson’s correlation coefficient. The threshold for significance was set at *p* < 0.05. All analyses were performed using the Prism 8 statistical software (ver. 8.4.3., GraphPad Software, San Diego, CA, USA).

## 3. Results

### 3.1. Tyr Enhanced Leu-Induced Muscle Anabolic Signaling and Muscular Hypertrophy in C2C12 Cells

To verify which amino acid enhanced Leu-induced anabolic signaling, murine C2C12 myoblasts were simultaneously exposed to Leu (2.5 mM) and other amino acids (2.5 mM) for 15 min. S6K phosphorylation (Thr 389) significantly increased in response to treatment with 2.5 mM Leu + 2.5 mM Tyr or Phe, compared with that in response to treatment with 5 mM Leu (Figure 1A). Both the Leu + Tyr and Leu + Phe treatments significantly increased S6K phosphorylation in a dose-dependent manner, although the Leu + Tyr treatment possibly induced a more significant effect (Figure 1B). Therefore, we investigated the booster effect of Tyr on Leu in subsequent experiments. At concentrations ranging from 0.25 to 2.5 mM, Tyr enhanced Leu-induced S6K phosphorylation in a dose-dependent manner (Figure 1C). Although Tyr alone exerted no effect on S6K phosphorylation, the Leu + Tyr combination induced significant S6K phosphorylation at a lower Leu concentration than when Leu was administered alone (Figure 1D,E). A similar trend was observed in the activation of other signaling pathways by mTORC1-related proteins, including 4E-BP, mTOR, and S6 (Figure 1F–I). The phosphorylation of proteins related to these pathways was mainly observed 10–30 min following amino acid exposure (Appendix A).

The SUnSET assay was used to detect incorporated puromycin in de novo proteins and demonstrated a significant increase in the protein synthesis rate in response to treatment with 5 mM Leu and the Leu + Tyr combination, i.e., 1 mM Leu + 0.5 mM Tyr and 4 mM Leu + 0.5 mM Tyr (Figure 2A,B). C2C12 cells were found to differentiate in the presence of Leu and/or Tyr, and the diameter of the myotube short axis and total protein content were quantified on day 5 post-differentiation (Figure 2C). Myotube diameter increased following the addition of amino acids, particularly following the addition of 0.5 mM Tyr to 1 mM Leu (Figure 2D). The total protein quantity significantly increased in the Leu and Tyr combination group versus in the control and 1 mM Leu groups (Figure 2E).

### 3.2. Tyr Enhanced Leu-Induced Muscle Anabolic Signaling in Isolated Muscles and Muscles of Orally Treated Mice

To determine whether the booster effect of Tyr on Leu could be observed ex vivo at physiological concentrations, we incubated isolated mouse soleus and EDL muscles with the KRB buffer supplemented with Leu and/or Tyr and then measured S6K and 4E-BP phosphorylation levels. The effect of the Leu and Tyr combination on S6K phosphorylation was similar to that induced by 3 mM Leu in both the soleus and EDL muscles (Figure 3A,B). Of note, the 1 mM Leu + 0.5 mM Tyr combination significantly promoted S6K phosphorylation in the soleus muscle as compared to 1 mM Leu (Figure 3A). In both cases, the extent of S6K phosphorylation induced by 0.5 mM and 1 mM Tyr in combination with 1 mM Leu was similar, with no dose dependence observed. 4E-BP phosphorylation showed the same trend as S6K phosphorylation, but to a lesser extent (Figure 3C,D).

Next, we evaluated the effect of oral Leu administration in combination with Tyr on S6K phosphorylation in the muscles. First, we determined the maximum dose of Leu employed in previous studies, and this was equivalent to the daily intake dose [23,24]. To ensure appropriate dosing, amino acids were dispersed in a 10% lipid emulsion and then administered to mice. Thirty minutes after amino acid administration, S6K phosphorylation in the soleus muscle was significantly higher in the Leu- (at a Leu dose of 10 mmol/kg BW) and Leu + Tyr-treated groups (both at doses of 5 mmol/kg BW Leu + 1 mmol/kg BW Tyr and 5 mmol/kg BW Leu + 5 mmol/kg BW Tyr) than in the control group (Figure 4A). S6K phosphorylation in the gastrocnemius muscle was significantly higher in the 10 mmol/kg BW Leu-treated groups and in the 5 mmol/kg BW Leu + 5 mmol/kg BW Tyr-treated groups than in the control group (Figure 4B). In addition, we preliminarily confirmed that S6K phosphorylation levels remained unchanged 15 and 60 min after amino acid administration. Serum Leu and Tyr concentrations increased in a dose-dependent manner 30 min after oral amino acid administration (Figure 4C,D). Unexpectedly, serum Tyr concentrations in the 5 mmol/kg BW Leu + 5 mmol/kg BW Tyr-treated groups were significantly higher than those in the 5 mmol/kg BW Tyr-treated groups (Figure 4D). The serum Tyr concentration was lower than that of Leu even when equal molar amounts of both amino acids were administered (Figure 4C,D). Although the oral bioavailability of Tyr was lower than that of Leu, probably due to its low water solubility, serum Tyr concentrations were positively correlated with S6K phosphorylation in both the soleus and gastrocnemius muscles in the Leu + Tyr-treated groups (Figure 4E–H); this was not the case with Leu.

### 3.3. Molecular Mechanism by Which Tyr Enhances Leu-Induced S6K Phosphorylation

To provide mechanistic insights into the booster effect of Tyr on Leu, first, we hypothesized that the incorporation of Leu into muscle cells is enhanced in presence of Tyr, and then we measured free Leu and Tyr levels in treated C2C12 cells and isolated muscles. However, we found that although Tyr was more significantly incorporated into cells than Leu at the same exposure concentrations, Leu incorporation remained unchanged irrespective of Tyr administration (Appendix A). 

Recent studies have shown that Sestrin1 and 2 bind to Leu and are involved in Leu-induced activation of mTORC1 signaling [14,18]. We compared the protein expression levels of Sestrin1 and 2 between C2C12 myoblasts, EDL muscles, gastrocnemius muscles, and soleus muscles. Both Sestrin1 and Sestrin2 were more highly expressed in C2C12 cells and soleus muscles than in EDL and gastrocnemius muscles (Figure 5A). Following the suppression of Sestrin1 expression in C2C12 cells through siRNA transfection, Sestrin2 expression was upregulated; however, following the suppression of Sestrin2 expression, Sestrin1 expression slightly decreased (Figure 5B). Thus, we examined S6K phosphorylation in C2C12 cells treated with Leu and/or Tyr under both Sestrin1 and 2 suppression. As Sestrin1 and 2 inhibit mTORC1 by interacting with GATOR2 [14,18], we expected that the suppression of the expression of these proteins would increase the basal S6K phosphorylation state and attenuate Leu-induced S6K phosphorylation. However, both the basal and Leu-induced S6K phosphorylation levels remained unchanged following the suppression of the expression of these proteins; S6K phosphorylation was found to be promoted by treatment with Tyr and was further enhanced following treatment with Leu in combination with Tyr (Figure 5C). Leucyl-tRNA synthetase (LRS) has been proposed as an intracellular Leu sensor that induces mTORC1 activation [28]. However, Leu and/or Tyr-induced S6K phosphorylation remained unchanged following LRS knockdown (Appendix A).

## 4. Discussion

In this study, we demonstrated that Tyr enhances Leu-induced mTORC1 signaling activation; however, it exerted no such effect when administered alone. This phenomenon was confirmed through both in vitro and in vivo experiments. To the best of our knowledge, this is the first study to elucidate the contribution of Tyr to mTORC1 signaling.

Leu is unique among EAA in that it stimulates anabolic signaling in muscles through the phosphorylation of mTOR, 4E-BP, and S6K [6]. The findings of previous in vitro studies on C2C12 cells have suggested that the Leu concentration threshold is approximately 5 mM, and that above this threshold, it induces significant levels of S6K phosphorylation, which is necessary for increased protein accretion in myotubes [29,30]. However, as only plasma Leu concentrations of approximately 1.5 mM have been observed in human subjects after the ingestion of high Leu doses (approximately 9 g) [31], a Leu concentration of 5 mM is assumed to be beyond physiological concentrations. The transient anabolic response induced by treatment with 1 mM Leu + 0.5 mM Tyr, which falls within the upper physiological concentration range, was equivalent to or greater than that induced by 5 mM Leu (Figure 1E–I). Our data indicated that the combination of Leu and Tyr at the appropriate ratio lowers the threshold for Leu-induced S6K phosphorylation to a level that falls within the physiological concentration range. The dose–response curves for Leu concentration versus phospho-S6K also supported this change in threshold (Figure 1C). Continuous Leu supplementation during differentiation increased myotube diameter but did not affect total protein levels (Figure 2D,E). Our findings are consistent with those of previous studies that showed that Leu preferentially induces myofibrillar protein synthesis [30,32]. However, continuous supplementation with Leu and Tyr significantly increased both myotube diameter and total protein levels, suggesting that cytoplasmic protein levels may also increase following Tyr-induced translation signaling enhancement.

The booster effect of Tyr on Leu-induced anabolic signaling activation was also confirmed in isolated muscles, and the extent of S6K and 4E-BP phosphorylation induced by the 1 mM Leu + 0.5 mM Tyr treatment was found to be equal to or greater than that induced by the 3 mM Leu treatment, especially in the soleus muscle (Figure 3). The intensity of the effect observed after incubation with 3 mM Leu was equivalent to that observed in mice with serum concentrations resulting from the ingestion of the maximum Leu dose of 10 mmol/kg BW (Figure 4C). In addition, S6K and 4E-BP phosphorylation seemed to be highest at Tyr concentrations of 0.5 mM and 1 mM, respectively, in combination with 1 mM Leu. Measurement of free Leu and Tyr intracellular levels showed that Tyr is more readily taken up by muscle cells than Leu (Appendix A), suggesting that high Tyr concentrations may not contribute in further promoting the anabolic signaling response. Our findings suggested that the delivery of Leu and Tyr to muscles at the appropriate ratio is important for inducing this booster effect, and that blood Leu and Tyr concentrations of 1 and 0.5 mM, respectively, may be optimal.

Thirty minutes after oral administration of 5 mmol/kg BW Leu + 5 mmol/kg BW Tyr, serum Leu and Tyr concentrations reached approximately 1.5 and 0.5 mM, respectively, and induced significant S6K phosphorylation in both the soleus and gastrocnemius muscles (Figure 4A,B). Serum Tyr concentrations following the oral administration of equal Tyr doses significantly increased upon the co-administration of Leu and Tyr (Figure 4D). However, further investigation is required to determine whether the simultaneous presence of Leu and Tyr in the intestinal tract can truly affect Tyr absorption. Although there was a variation in serum Tyr levels, probably due to its low water solubility and low bioavailability, these levels were positively correlated with S6K phosphorylation levels (Figure 4E,F). Our data suggest that postprandial blood Tyr concentration is in an appropriate ratio with that of Leu, a key determinant of muscle anabolic signaling.

Notably, we found that the inhibition of Sestrin1 and 2 expression did not affect the Leu-induced S6K phosphorylation (Figure 5C); however, it is possible that the knockdown level was insufficient, and the results could have been different if the extent of knockdown (or knockout) had been greater. As multiple intracellular Leu sensors, such as LRS, SAR1B, and perhaps other unknown factors, have been identified or proposed [28,33,34,35], their expression may be upregulated in a compensatory manner, and this might maintain Leu-induced S6K phosphorylation under Sestrin1 and 2 suppression. The inhibition of LRS expression did not affect Leu- and/or Tyr-induced S6K phosphorylation (Appendix A). LRS was identified in HEK293 cells [28]; however, inhibition of its expression in C2C12 cells suppressed muscle differentiation but did not affect muscle hypertrophy [36]. These findings suggest that LRS may contribute little to the regulation of mTORC1 activity in skeletal muscles. Another Leu sensor, SAR1B, is highly expressed in skeletal muscles [33]. However, the dissociation constant of SAR1B from Leu was lower than that of the Sestrins from Leu. SAR1B is thought to respond to low Leu levels to maintain basal mTOR activity. Thus, the functional relationship between Leu sensors and mTORC1 activity in the muscle cells should be explored in future studies. Tyr can induce S6K phosphorylation under Sestrin1 and 2 suppression (Figure 5C). In the presence of Leu, the interaction between Sestrins and GATOR2 is disrupted through the binding of Leu to Sestrins [14,18]. Leu-dependent Sestrin divergence conditions are analogous to the state of inhibition of Sestrin expression, and in this state, Tyr can activate mTORC1 via an unknown interaction with GATOR2. As Phe also exerted a booster effect on Leu-induced S6K phosphorylation (Figure 1A,B), the structural features of both Tyr and Phe are probably involved in the molecular mechanism; however, the precise mechanism by which Tyr (and Phe) promotes mTORC1 activation in the presence of Leu is yet to be elucidated.

We found the protein expression levels of Sestrin1 and 2 in the soleus muscle to be higher than that in the EDL and gastrocnemius muscles (Figure 5A); this might explain the more significant booster effect of Tyr on Leu-induced mTORC1 activation in the soleus muscle (Figure 3 and Figure 4). However, this speculation is inconsistent with in vitro findings, as the booster effect of Tyr on Leu was found to be significant under Sestrin1 and 2 suppression (Figure 5C). Recent studies have shown that Sestrin1 levels decrease with lack of use or aging, and that it is a key regulator of anabolic and degradative pathways that prevent muscle atrophy [37,38,39]. Sestrin1 is upregulated following acute resistance exercise [40] and downregulated following chronic treadmill exercise [41]. Although the relationship between the change in Sestrin expression and mTORC1 activity is not clearly understood, and as changes in the expression of other confounding factors that affect mTORC1 activity need to also be considered in this regard, the expression levels of Sestrins could partially explain the upregulation in Leu (and Tyr)-induced mTORC1 activation with increased MPS following resistance exercise- or age-related anabolic resistance.

Finally, this study has some limitations. First, in our in vivo experiments, we evaluated only S6K phosphorylation and did not evaluate the actual rate of MPS. Second, the acute anabolic response level, including S6K phosphorylation and MPS rate, is not sufficient to estimate the divergence of chronic intervention-induced changes in muscular hypertrophy [42]. Although we demonstrated that Tyr serves as a booster to enhance Leu-induced MPS and muscular hypertrophy in vitro (Figure 2), further studies need to be carried out to determine whether long-term nutritional intervention with Leu and Tyr, or a protein source that contains them at high levels, could result in muscular hypertrophy in vivo. Other recent studies have reported that Leu, isoleucine, and valine (BCAAs) in combination with alanine at specific ratios enhance BCAA bioavailability and promote anabolic responses in the skeletal muscles [43,44]. Thus, the most appropriate amino acid composition for achieving optimal nutritional benefits should be validated in future studies.

## 5. Conclusions

In conclusion, our findings demonstrate that Tyr enhances Leu-induced activation of mTORC1 signaling in C2C12 cells. A similar effect was observed in isolated murine muscle and the skeletal muscles of mice orally administered the amino acids. Our findings may provide important factors to be considered when establishing amino acid combinations or blends of various animal- and plant-based proteins suitable for promoting muscle synthesis [45] and may help in the development of nutritional approaches for preventing and treating sarcopenia and frailty.

## 6. Patents

The Kao Corporation has filed patents to which the outcome of this research is relevant (JP6496599B2).

## Figures and Tables

**Figure 1 nutrients-16-00084-f001:**
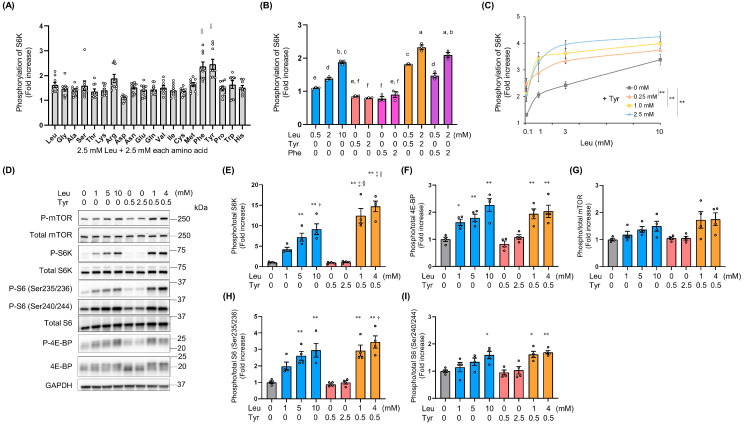
Tyrosine enhances leucine-induced S6K phosphorylation in C2C12 myoblasts. (**A**) S6K phosphorylation (Thr389) after 15 min of stimulation with 2.5 mM Leu and 2.5 mM of each amino acid (*n* = 9) as detected through the alpha screen assay. (**B**) S6K phosphorylation in response to treatment with Leu and/or Tyr or Phenylalanine (Phe) (*n* = 3) as detected through the alpha screen assay. (**C**) Dose–response evaluation of S6K phosphorylation at Leu and Tyr concentrations ranging from 0 to 10 mM and from 0 to 2.5 mM (*n* = 3), respectively, as detected through the alpha screen assay. (**D**) Representative images of anabolic signaling pathways detected in myoblasts on day 1 post-differentiation through western blotting following 15 min of stimulation with Leu and/or Tyr. Phosphorylation ratios for S6K (Thr 389) (**E**), 4E-BP (Thr37/46) (**F**), mTOR (Ser2448) (**G**), and S6 (Ser235/236, Ser 240/244) (**H**,**I**) were calculated by dividing the phosphorylation levels by the protein expression levels (*n* = 4). All data represent the fold change with respect to the control (0 mM Leu and 0 mM Tyr). Data are presented as mean ± SEM. Circles represent individual values. * *p* < 0.05, ** *p* < 0.01 vs. control † *p* < 0.05, ‡ *p* < 0.01 vs. 1 mM Leu. § *p* < 0.05, || *p* < 0.01 vs. 5 mM Leu as determined by one-way ANOVA (**A**,**E**–**I**). Different letters (a–f) represent significant differences (*p* < 0.05) as determined by one-way ANOVA (**B**). ** *p* < 0.01 vs. the Leu alone condition as determined by two-way ANOVA (**C**).

**Figure 2 nutrients-16-00084-f002:**
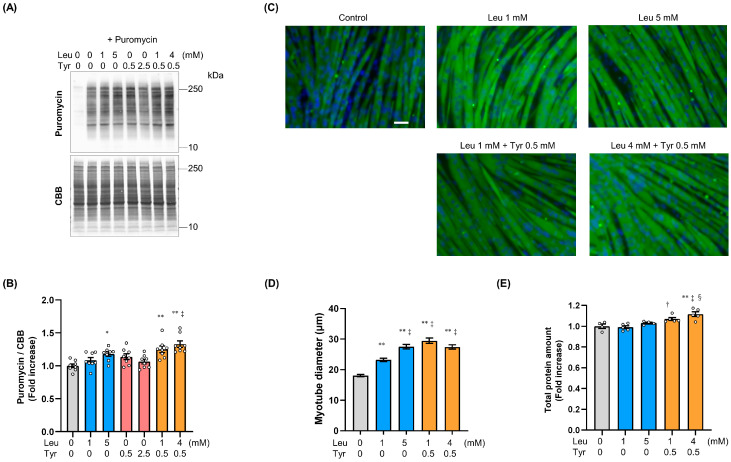
Tyrosine enhances leucine-induced muscle protein synthesis and myotube hypertrophy in C2C12 cells. Muscle protein synthesis was assessed through the surface sensing of translation (SUnSET) method. (**A**) Representative images of puromycin incorporation in myoblasts on day 1 post-differentiation as detected through western blotting and CBB staining. (**B**) The rate of puromycin incorporation was normalized to the corresponding CBB staining (*n* = 8). Differentiating C2C12 myoblasts were continuously treated with Leu and/or Tyr during differentiation. (**C**) Representative images of myotubes stained with MHC (green) and DAPI (blue) on day 5 post-differentiation (scale bar: 50 µm) (**D**) and myotube diameters (*n* = 133–142). (**E**) Total protein levels were evaluated through the BCA analysis on day 5 post-differentiation (*n* = 4–5). All data represent the fold change with respect to the control (0 mM Leu and 0 mM Tyr). Data are presented as mean ± SEM. Circles represent individual values. * *p* < 0.05, ** *p* < 0.01 vs. control. † *p* < 0.05, ‡ *p* < 0.01 vs. 1 mM Leu. § *p* < 0.05 vs. 5 mM Leu as determined by one-way ANOVA.

**Figure 3 nutrients-16-00084-f003:**
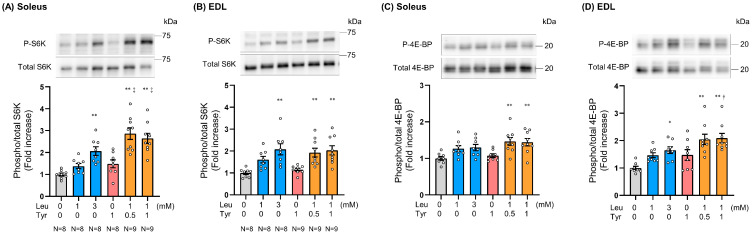
Effect of leucine and tyrosine on S6K and 4E-BP phosphorylation in isolated muscles. Overnight-fasted male C57BL/6J mice were sacrificed, and their soleus and extensor digitorum longus (EDL) muscles were isolated and incubated in Krebs Ringer Bicarbonate (KRB) buffer containing 0–3 mM Leu and/or 0–1 mM Tyr for 20 min. (**A**,**B**) S6K phosphorylation (Thr389) levels in the soleus and EDL muscles. (**C**,**D**) 4E-BP phosphorylation (Thr37/46) levels in the soleus and EDL muscles. All data represent the fold change with respect to the control (0 mM Leu and 0 mM Tyr). Data are presented as mean ± SEM (*n* = 8–9). Circles represent individual values. The number of samples in each group is indicated under the bar charts. * *p* < 0.05, ** *p* < 0.01 vs. control. † *p* < 0.05, ‡ *p* < 0.01 vs. 1 mM Leu as determined by one-way ANOVA.

**Figure 4 nutrients-16-00084-f004:**
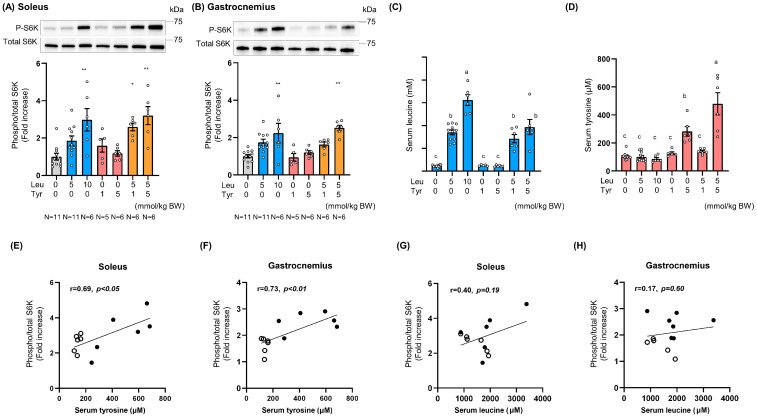
Effect of oral leucine and tyrosine administration on S6K phosphorylation in mice. A lipid emulsion containing Leu and/or Tyr was intragastrically administered to overnight-fasted mice under isoflurane anesthesia. Muscle and blood samples were collected after 30 min. S6K phosphorylation (Thr389) in the soleus (**A**) and gastrocnemius (**B**) muscles was detected by western blotting. Serum leucine (**C**) and tyrosine (**D**) levels were determined through LC-MS/MS. (**E**–**H**) The correlation between S6K phosphorylation level and serum tyrosine or leucine concentration in the Leu + Tyr-treated groups is shown. Empty circles with a black outline and black circles indicate the 5 mmol/kg BW Leu + 1 mmol/kg BW Tyr and 5 mmol/kg BW Leu + 5 mmol/kg BW Tyr treatments, respectively. All data represent the fold change with respect to the control (0 mmol/kg BW Leu + 0 mmol/kg BW Tyr). Data are presented as mean ± SEM (*n* = 5–11). Circles represent individual values. The number of samples in each group is indicated under the bar charts. * *p* < 0.05, ** *p* < 0.01 vs. control as determined by one-way ANOVA (**A**,**B**). Different letters (a–c) represent significant differences (*p* < 0.05) as determined by one-way ANOVA (**C**,**D**). A correlation analysis was performed using Pearson’s correlation coefficient (**E**–**H**).

**Figure 5 nutrients-16-00084-f005:**
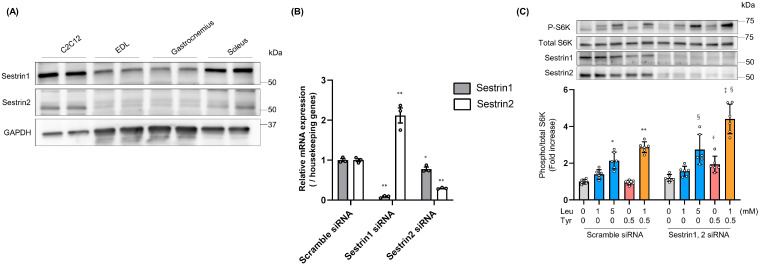
Involvement of Sestrin1 and 2 in leucine- and tyrosine-induced S6K phosphorylation. (**A**) Sestrin1 and 2 expression in mouse C2C12 myoblasts, EDL muscles, gastrocnemius muscles, and soleus muscles was determined through western blotting. (**B**) On day 1 post-differentiation, Sestrin1 and 2 mRNA expression levels in C2C12 myoblasts subjected to siRNA transfection were determined via qRT-PCR (*n* = 3). (**C**) S6K phosphorylation (Thr389) in C2C12 myoblasts on day 1 post-differentiation in response to 15 min of stimulation with Leu and/or Tyr following the downregulation of Sestrin1 and 2 expression using siRNA (*n* = 6). All data represent the fold change with respect to the scramble control (Scr) siRNA (0 mM Leu and 0 mM Tyr). Data are presented as mean ± SEM. Circles represent individual values. * *p* < 0.05, ** *p* < 0.01 vs. control (Scr siRNA). † *p* < 0.05 vs. 0.5 mM Tyr (Scr siRNA). ‡ *p* < 0.01 vs. 1 mM Leu + 0.5 mM Tyr (Scr siRNA). § *p* < 0.01 vs. control (Sestrin1, 2 siRNA) as determined by two-way ANOVA.

## Data Availability

The datasets used or analyzed during the current study are available from the corresponding author upon reasonable request.

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
