# Peer review of "Tyrosine Is a Booster of Leucine-Induced Muscle Anabolic Response"

_nutrients, 2023, doi:10.3390/nu16010084_

Round 1

Reviewer 1 Report

Comments and Suggestions for Authors

The manuscript describes some effects of non-essential amino acid tyrosine on leucine-induced activation of the mTORC1 signaling pathway in C2C12 cells, in isolated muscles and in vivo. The authors of the manuscript showed that that tyrosine is able to enhance the leucine-induced activation of mTORC1 signaling, however the use of tyrosine alone has no such effect on anabolic signaling. While the manuscript is clearly written and the reported findings are interesting, there are some concerns that need to be addressed before publication of the manuscript.

Major comments

1. Using the SUnSET method (Goodman et al. // FASEB J. 2011;25(3):1028-39. doi: 10.1096/fj.10-168799), it is important to measure the rates of muscle protein synthesis in both isolated muscles (ex vivo) and in vivo (after oral administration of leucine and tyrosine). It is critical since an assessment of phosphorylation status of the mTORC1 substrates per se may not reflect the actual rates of protein synthesis in muscle tissue. There are mTORC1-independent pathways that can be involved in the regulation of translation initiaion, elongation and ribosome biogenesis.

2. Using a single housekeeping gene for RT-PCR is no longer considered as enough for years, even though many articles are still published with only one reference gene. Two housekeeping genes is the minimum and using 3 of them is better. 

Minor comment

1. The manuscript would benefit from proofreading by a native English speaker.

Comments on the Quality of English Language

The manuscript should be proofread and checked for grammar errors/stylistics.

Reviewer 2 Report

Comments and Suggestions for Authors

In the present study, the Authors – Tamura et al. – aimed at providing new insight into the potential of amino acid mixtures to stimulate muscle anabolism and, particularly, to identify the best amino acid to be combined with leucine, EAA known to stimulate the mTOR pathway, to possibly boost its action on skeletal muscle. Their work provides interesting evidence about the ability of tyrosine to enhance leucine-induced activation of mTORC1 signaling, both in vivo and in vitro at specific ratios. The manuscript is generally well-written, clearly stating the objectives of their research, as well as the limitations encountered during their research. Nonetheless, I have some considerations that are listed below.

MATERIALS AND METHODS

-    This section should be divided in numbered paragraphs as the other sections of the paper.

-      Line 120: If I understand correctly, a cohort of C57BL/6 male mice was exclusively used to perform in vitro incubation of isolated muscles (soleus, EDL) with KRB buffer 133 supplemented with leucine and/or tyrosine, whilst different groups of BL6 mice were assigned to the in vivo supplementation with vehicle or amino acids. So, how were mice groups organized? How many animals per group?

Why did the Authors choose to use mice between 7 and 14 weeks of age and not all age-matched?

A scheme to detail study design would be very helpful for the reader.

RESULTS

-    Figure quality and size should both be improved, since it is difficult to appreciate details in image panels. Also, for mice experiments, the n of each group should be indicated in the legend or caption.

-    Correct the typo “phosho” on the Y axis in WB graphs for S6K and 4E-BP.

DISCUSSION

-    Lines 400-405: As described by the Authors, finding the appropriate ratio at which nutritional supplements should be administered is indeed crucial, and excessive amounts of leucine are known to induce hypoglycemia and to increase ammonia levels in the blood. Maybe the Authors should mention that the importance of identifying the best ratio for EAAs – based supplements, has been described in recent literature, e.g., it has been demonstrated that the anabolic effect of leucine, often associated with isoleucine and valine (i.e. BCAAs), in ratio 2:1:1, can be effectively boosted by a combination with two equivalents of alanine, one of the amino acids involved in BCAAs catabolism, in physiological and aging muscle conditions (Mantuano et al., Nutrients 2020 doi: 10.3390/nu12082295, and 2023 doi: 10.3390/nu15020330).

Round 2

Reviewer 1 Report

Comments and Suggestions for Authors

Although I am generally satisfied with the answers provided by the authors, all conclusions concerning changes in protein synthesis should be revised since the authors did not measure the actual rates of protein synthesis.  For example, lines 409-410 ("muscle protein synthesis" can be replaced by "mTORC1-signaling"), line 509 ("and protein synthesis" should be removed).

Author Response

We appreciate the reviewer’s careful reading of our manuscript and fully acknowledge the comment. As indicated, we revised the manuscript in line 408 and line 507.